# Breast self-examination practice and its determinants among women in Ethiopia: A systematic review and meta-analysis

Yordanos Gizachew Yeshitila[1]*, Getachew Mullu Kassa[2], Selamawit Gebeyehu[3‡], Peter Memiah[4‡], Melaku Desta[5]

1 School of Nursing, College of Medicine and Health Science, Arba Minch University, Arba Minch, Ethiopia, 2 College of Health Sciences, Debre Markos University, Debre Markos, Ethiopia, 3 School of Public Health, College of Medicine and Health Science, Arba Minch University, Arba Minch, Ethiopia, 4 Division of Epidemiology and Prevention: Institute of Human Virology, University of Maryland School of Medicine, Baltimore, Maryland, United States of America, 5 Department of Midwifery, College of Health Science, Debre Markos University, Debre Markos, Ethiopia

☯ These authors contributed equally to this work.
‡ SG and PM also contributed equally to this work.
* yordanos.gizachew@yahoo.com

**Data Availability Statement:** All relevant data are within the paper and its Supporting Information files.

## Abstract

### Background

The survival rate from breast cancer is lowest in African countries and the distribution of breast self-examination practice of and its determinants are not well investigated in Ethiopia. Therefore, this systematic review and meta-analysis was designed to determine the pooled prevalence of breast self-examination and its associated factors among women in Ethiopia.

### Methods

Preferred Reporting Items for Systematic Reviews and Meta-Analyses (PRISMA) guideline was followed for this systematic review and meta-analysis. The databases used were; PUBMED, Cochrane Library, Google Scholar, CINAHL, African Journals Online, Dimensions and Summon per country online databases. Search terms used were; breast self-examination, breast cancer screening, early detection of breast cancer and Ethiopia. Joanna Briggs Institute Meta-Analysis of Statistics Assessment and Review Instrument (JBI-MAStARI) was used for critical appraisal of studies. The meta-analysis was conducted using STATA 15 software. The pooled meta-analysis was computed to present the pooled prevalence and relative risks (RRs) of the determinate factors with 95% confidence intervals (CIs).

### Results

We identified 2,637 studies, of which, 40 articles (with 17,820 participants) were eligible for inclusion in the final meta-analysis. The pooled estimate of breast self-examination in Ethiopia was 36.72% (95% CI: 29.90, 43.53). The regional distribution breast self-examination ranged from 21.2% (95% CI: 4.49, 37.91) in Tigray to 61.5% (95% CI: 53.98, 69.02) in

**Funding:** The author(s) received no specific funding for this work.

**Competing interests:** The authors have declared that no competing interests exist.

**Abbreviations:** BSA, breast self-awareness; CI, Confidence Interval; NCDs, Non Communicable Disease; GLOBOCAN, Global Cancer Incidence, Mortality and Prevalence; OR, odds ratio; SNNPR, Southern Nations, Nationalities, and People's Region; WHO, World Health Organization.

Gambela region. The lowest prevalence of breast self-examination was observed among the general population (20.43% (95% CI: 14.13, 26.72)). Women who had non-formal educational status (OR = 0.4 (95% CI: 0.21, 0.77)), family history of breast cancer (OR = 2.04 (95% CI: 1.23, 3.39)), good knowledge of breast self-examination (OR = 4.8 (95% CI: 3.03, 7.6)) and favorable attitude toward breast self-examination (OR = 2.75, (95% CI: 1.66, 4.55)) were significantly associated with practice of breast self-examination.

## Conclusions

Only a third of women in Ethiopia practiced breast examination despite WHO guidelines advocating for this practice among all women of reproductive age. Intervention programs should address the factors that are associated with breast self-examination. Population specific programs are needed to promote breast self-examination.

## Introduction

In the 21st century, cancer is expected to rank as the leading cause of death and the single most important barrier to increasing life expectancy in every country of the world [1, 2]. Female breast cancer is the second leading cause of death amongst all cancers (11.6%) and the leading cause of death amongst women. Breast cancer is also the most commonly diagnosed cancer among females. According to Global Cancer Incidence, Mortality and Prevalence (GLOBOCAN) 2018 estimates, there were about 2.1 million newly diagnosed female breast cancer cases in 2018, accounting for almost 1 in 4 cancer cases among women [2]. The condition is the most frequently diagnosed cancer in the vast majority of the countries, however Sub-Saharan Africa has a higher increase of cervical cancer diagnosis [2–4].

World Health Organization (WHO) recommends cancer prevention as essential component of all cancer control plans because about 40% of all cancer deaths can be prevented [5]. Breast cancer is a common cause of cancer morbidity and mortality in women. Breast self-examination (BSE) has been promoted for many years as screening method for breast cancer at an early stage, in order to decrease the risk of dying from breast cancer [6].

In Ethiopia, a National Cancer Control plan was established in 2015 with an objective to promote cancer prevention and early detection, and improve diagnosis and treatment, including palliative care [7]. Cancer accounts for about 5.8% of total national mortality in Ethiopia. Breast cancer accounts for 30.2% of all cancers among the adult population in the country. About two-thirds of annual cancer deaths occur among women [8]. One of the main reasons cited for the high mortality from cancer is the late presentation of women with advanced breast cancer disease [9].

In line with the World Health Organization (WHO) global cancer control strategy, the Ethiopian cancer control program planned to reach half of the country's population with cancer prevention awareness information by 2020. The plan was owing to the high burden of mortality from breast cancer, which accounts for 34% of all total cancer cases. In low and middle-income countries, where the disease is diagnosed in late stages and resources are very limited, early diagnosis remains an important detection strategy especially in Low- and Middle-Income Countries (LMIC) where resources are limited and the disease is diagnosed in the late stages. There is evidence that the practice of breast self-examination empowers women, an important first step to encourage women to actively be responsible for their own health, especially for

those in LMIC with limited resources and access to other forms of preventive healthcare [10]. Breast self-examination enables women to be aware of any changes in their breast and provides them with some acknowledgement of the part they can play in being empowered to fight breast disease [11].

The survival rate from cancer was lowest in African countries, with poorly developed health services, as indicated by limited availability of cancer diagnostic and treatment facilities [12, 13]. Notwithstanding, the decrease in breast cancer mortality in Africa has been attributed with increased breast cancer awareness [14].

The practice of BSE varies across Ethiopia with different variations among women [15–24]. There are several determinant factors that have been identified for breast self-examination practice such as educational status of women, knowledge about breast self-examination and breast cancer, attitude towards BSE, family history of breast cancer, perceived susceptibility to breast cancer and other sociodemographic variables. Despite the national response to the burden of cancer in line with the United Nations High-Level Meeting on the Control of NCDs (Non Communicable Disease) and the Global Action Plan for the Control of NCDs 2013–2020, the burden and mortality from breast cancer is still high among women in Ethiopia [8, 25, 26].

Though there is sufficient evidence in this area, the study results are variable, and therefore it is practical to work towards synthesizing the evidence. The aim of this systematic review and meta-analysis was, therefore, to estimate the pooled prevalence of breast self-examination practice and associated factors among women in Ethiopia. The findings of this study will help policy makers and other stakeholders in planning and implementing strategies to enhance uptake of breast self-examination practice and reduce the incidence of breast cancer.

## Methods

### Systematic review registration and reporting of findings

The protocol has been registered on an International Prospective Register of Systematic Review (PROSPERO), University of York Center for Reviews and Dissemination https://www.crd.york.ac.uk/CRDWeb/ registration number CRD42020190224. The findings of the review was reported in accordance to the recommendation of the Preferred Reporting Items for Systematic Review and Meta-Analysis (PRISMA-P) 2009 statement guideline (S1 Checklist) [27].

### Study design and search strategy

Published and unpublished research articles that were conducted to assess breast self-examination practice in Ethiopia were included in this systematic review and meta-analysis. An intensive search was done from PMC/MEDLINE, CINAHL, Cochrane, Global index medicus, African Journals Online databases, Dimensions and Summon per country online databases to access articles done on practice of breast self-examination. Besides, we also conducted direct hand searches on Google and Google scholar to retrieve additional articles. Moreover, reference lists of screened studies were checked. The search was administered by two authors (YGY and MDA) independently. The term 'Breast self-examination' was searched with all of the subsequent terms as a mix of free text and thesaurus terms in numerous variations like breast cancer screening, early detection of cancer, breast examination, practice and Ethiopia. Moreover, the subsequent keywords were familiarized to retrieve studies from PMC database (S1 File). Studies that were relevant after title and abstract screening were assessed by full text to see those that provided adequate data to be included in this study.

## Study selection and eligibility criteria

**Eligibility criteria.** *Inclusion criteria*. Articles reporting the prevalence of breast self-examination practice and predictors among women in Ethiopia were included in this study. In particular, studies that were included in this study considered the following criteria:

*Study area*. only studies conducted in Ethiopia.

*Study design*. All observational studies (cross-sectional, case controls, and cohort) that reported original data on the prevalence of breast self-examination practice and its predictors among women in Ethiopia were considered.

*Language*. articles written in English language.

*Population*. Studies that were conducted among women in Ethiopia.

*Publication condition*. both published articles and unpublished research were included in this study.

Exclusion criteria.

- Studies which did not report the outcome variable of this review.

- Studies which did not specify study population and qualitative studies.

- Non-accessible research which were unpublished, irretrievable from the databases or failed to receive replies to email from corresponding authors were excluded.

- Duplicate publications

## Outcome of the study

The main outcome of this study is the practice of breast self-examination. It is defined as a process whereby women examine their breasts regularly to detect any abnormal swelling or lumps in order to learn the topography of her breast [28], to know how her breasts normally feel and be able to identify changes in the breast should they occur in the future [29]. Regular breast self-examination is when breast self-examination is performed each month at the same time after some day menstrual cycle [30, 31]. Occasional breast self-examination is when BSE is 1 to 3 times a year or every 3 months (irregularly at any time) [30]. This study also assessed the association between selected independent variables with breast self-examination practice. The independent variables included in this study were educational status, family history of breast cancer, knowledge towards breast self-examination and attitude towards breast self-examination.

## Data extraction and quality assessment

The primary online search for the review was conducted between February 16 to May 21 2020. We have included studies irrespective of their publication date through May 21 2020. The review followed PRISMA flow chart to identify and select relevant studies for this analysis. Primarily, duplicated retrievals were removed. Then, studies whose titles were irrelevant for this study were excluded. After that, the abstracts were assessed and screened based on the outcome variables. At this stage, studies that were not relevant for this study were excluded. For the remaining articles, the full text was assessed. The eligibility of these articles was assessed based on the pre-set inclusion criteria. When articles did not have sufficient data, corresponding authors of the research articles were contacted.

Then, the two reviewers (YGY and MDA) independently assessed or extracted the articles for overall quality and or inclusion in the review using a standardized data extraction format. The third author (GMK) reconciled any disagreements that occurred among the two authors.

From each study, we extracted data on the study location, region, publication year, study design, sample size, and author name for the overall practice of breast self-examination (Table 1).

The quality of included studies was assessed in accordance with the relevant critical Joanna Brigs Institute (JBI) appraisal checklist (JBI). Two independent reviewers (YGY & MDA) did a full-text quality assessment. Then, a combined quality score ranging from 0 to 9 was assigned to each included article (S2 File). Finally, studies with the scale of $\geq$ 6 out of 9 were categorized as high quality [32].

## Publication bias and statistical analysis

The publication bias was assessed using Egger's tests. A p-value of less than 0.05 was used to declare a statistically significant presence of publication bias. $I^2$ test statistics were used to investigate the presence of heterogeneity across the included studies. The $I^2$ test statistics of 25, 50 and 75% was declared as low, moderate and high heterogeneity, respectively, and a p-value less than 0.05 was used to declare statistically significant heterogeneity. For test results with the presence of heterogeneity, a random effect model were used as a method of analysis.

Data were extracted in Microsoft Excel and then exported to STATA version 15 for further analysis. Forest plot was used to present the combined estimate with 95% confidence interval (CI) of the meta-analysis. Subgroup analysis was conducted by regions of the country and study population. A meta-regression model was done based on sample size and year of publication to identify the sources of random variations among included studies. The effect of selected determinant variables which include family history of breast illness, knowledge of breast cancer and attitude toward breast cancer were analyzed using separate categories of meta-analysis. The findings of the meta-analysis were presented using forest plot and Odds Ratio (OR) with its 95% Confidence Interval (CI).

## Results

### Study identification

A total of 2,637 research articles were identified by electronic search in PMC/MEDLINE, CINAHL, Cochrane, Global index medicus, Dimensions and Summon per country and African Journals Online data bases. Of which, 44 were excluded due to duplication, 2,493 through review of titles and abstracts. Additionally, four articles were excluded due to failed retrieval of full text and two full-text articles were excluded for not reporting the outcome variable. Finally, 40 studies were found to be eligible and included in the meta-analysis (Fig 1).

**Characteristics of included studies.** In the current systematic review and meta-analysis, two administrative towns and six regions in the country were represented. Most studies were conducted in Addis Ababa (n = 11) [33–43] and Oromia region (n = 9) [31, 44–51]; followed by South Nation Nationality of People (SNNP) (n = 8) [52–59], Amhara region (n = 6) [30, 60–64], Tigray region (n = 3) [65–67] and one study from Gambella region [68], one from Dire Dawa city administration [69] and one from Harari region [70]. The studies were conducted from 2011 to 2020. All of the studies were cross-sectional design. The sample size of the included studies ranged from 126 [70] to 845 [67] participants, which comprised a total of 17,820 study participants. The methodological quality was determined by the relevant critical JBI appraisal checklist (JBI) and the quality of the majority of studies 39 (97.5%) was high (Table 1). Out of the 40 articles included, 9 of them were unpublished [34, 37, 39–41, 51, 54, 56, 69].

**Table 1. Summary characteristics of studies included in study of breast self-examination among women in Ethiopia.**

| Authors | Year | Study setting | Study design | Study area, Region | Residency | Study Population | Age | Sample size | Prevalence | Quality |
|---|---|---|---|---|---|---|---|---|---|---|
| Abay M et al. [65] | 2018 | Facility based | CR | Adwa, Tigray | Urban | 20–70 Years old women | 20–70 | 404 | 6.5 | 8 |
| Abeje S et al. [33] | 2019 | Facility based | " | Addis Ababa | Urban | 20–49 Years old women | 20–49 | 633 | 24 | 9 |
| Aman S et al. [34] | Unpublished | Facility based | " | Addis Ababa | Urban | 20 years and older women | Not clearly stated | 422 | 32.2 | 8 |
| Getu MA A et al. [35] | 2019 | Institution based (school) | " | Addis Ababa | Urban | Students | Not clearly stated | 407 | 21.4 | 9 |
| Ameer K et al. [70] | 2014 | Facility based | " | Haromaya, Harari | Urban | Students | Not clearly stated | 126 | 23 | 5 |
| Azage M et al. [60] | 2013 | Facility based | " | West gojjam, Amhara | Urban | HEW | 15–44 | 403 | 37.3 | 9 |
| Getahun, B.S. [71] | 2011 | Institution based (school) | " | Gondor, Amhara | Urban | Employees | 19–56 | 403 | 46 | 8 |
| Bikila T et al. [36] | 2016 | Institution based (school) | " | Addis Ababa | Urban | Students | 15–29 | 662 | 43 | 9 |
| Birhane, K et al. [62] | 2017 | Institution based (school) | " | Debrebirehan Amhara | Urban | Students | Not clearly stated | 420 | 28.3 | 9 |
| Birhane, N et al. [52] | 2015 | Institution based (school) | " | Keffa zone, SNNPR | Urban and rural | Teachers | Not clearly stated | 316 | 12 | 9 |
| Chimid, C. et al. [37] | Unpublished | Community Based | " | Addis Ababa | Urban | 20–49 years older women | 20–49 | 630 | 39.6 | 9 |
| Dagnaw, M [63] | 2019 | Institution based (school) | " | Gondor, Amhara | Urban | Students | Not clearly stated | 806 | 45.8 | 9 |
| Dagne, AH et al. [64] | 2019 | Facility based | " | Debre Tabor, Amhara | Urban | Working in health facilities | 20–50 | 421 | 32.5 | 8 |
| Desta, F et al. [44] | 2018 | Institution based (school) | " | Jimma, Oromia | Urban | Students | 21–25 | 200 | 46.5 | 9 |
| Dibisa, T. M. et al. [45] | 2019 | Community Based | " | Kersa district, Oromia | Urban and rural | 20 years and older women | Not clearly stated | 422 | 3.6 | 9 |
| Dinegde NG et al. [38] | 2019 | Facility based (school) | " | Addis Ababa | Urban | Students | Not clearly stated | 381 | 24.6 | 7 |
| Agide FD et al. [53] | 2019 | Community Based | " | Hadiya zone, SNNPR | Urban and rural | RAW | 15–49 | 810 | 8.7 | 8 |
| Getie, A et al. [54] | Unpublished | Community Based | " | Arbaminch district, SNNPR | Urban and rural | 20–64 years older women | 20–64 | 634 | 21.3 | 7 |
| Hailu, T et al. [66] | 2016 | Institution based (school) | " | Mekelle, Tigray | Urban | Students | Not clearly stated | 792 | 37.2 | 9 |
| Jemebere, W [55] | 2019 | Facility based | " | Hawassa, SNNPR | Urban | Health care professionals | Not clearly stated | 180 | 71.2 | 8 |
| Amogne FK et al. [46] | 2014 | Institution based (school) | " | Bale Zone, Oromia | Urban and rural | Students | Not clearly stated | 422 | 15.5 | 9 |
| Kassa, RT et al. [47] | 2017 | Institution based (school) | " | Adama, Oromia | Urban | Students | Not clearly stated | 423 | 20 | 7 |
| Legesse and Gedif [67] | 2014 | Community Based | " | Mekelle, Tigray | Urban | 20 years and older | Not clearly stated | 845 | 20 | 9 |
| Lera, TA et al. [56] | Unpublished | Community Based | " | Wolayita, SNNPR | Urban | 20–65 years old women | 20–65 | 629 | 34.5 | 8 |
| Mekonen, M et al. [39] | Unpublished | Facility based | " | Addis Abeba | Urban | 20–70 years old women | 20–70 | 422 | 11.3 | 9 |
| Mekuria, M et al. [57] | 2020 | Institution based (school) | " | Gamo Gofa zone, SNNPR | Urban | Teachers | Not clearly stated | 247 | 34.3 | 8 |

*(Continued)*

**Table 1.** (Continued)

| Authors | Year | Study setting | Study design | Study area, Region | Residency | Study Population | Age | Sample size | Prevalence | Quality |
|---|---|---|---|---|---|---|---|---|---|---|
| Minasie A et al. [58] | 2017 | Facility based | " | Wolayita Zone, SNNPR | Urban and rural | HEW | 19–39 | 281 | 45.6 | 9 |
| Natae, S F [31] | 2015 | Institution based (school) | " | Ambo, Oromia | Urban | Students | Not clearly stated | 320 | 20.7 | 9 |
| Negeri, E L et al. [48] | 2017 | Facility based | " | Western Ethiopia, Oromia | Urban and rural | Health care professionals | 18–50 | 314 | 77 | 8 |
| Segni, MT et al. [49] | 2016 | Institution based (school) | " | Adama, Oromia | Urban | Students | Not clearly stated | 368 | 39.4 | 7 |
| W/Tsadik S et al. [40] | Unpublished | Facility based | " | Addis Abeba | Urban | Health care professionals | 20–60 | 422 | 71 | 9 |
| Shallo, S A et al. [50] | 2019 | Facility based | " | West shoa, Oromia | Urban | Health care professionals | 19–43 | 379 | 32.6 | 8 |
| Solomon F et al. [69] | Unpublished | Facility based | " | Dire Dawa | Urbna | 20 years and older women | 20–72 | 315 | 32 | 7 |
| Tewabe, T et al. [30] | 2019 | Institution based (school) | " | Bahirdar, Amhara | Urban | Students | 18–40 | 222 | 54 | 8 |
| Tewelde, B et al. [41] | Unpublished | Institution based (school) | " | Addis Abeba | Urban | Teachers | 22–59 | 589 | 43.6 | 9 |
| Wodera, AL et al. [51] | Unpublished | Community Based | " | Bale Zone, Oromia | Urban and rural | RAW | 15–49 | 841 | 13 | 6 |
| Wurjine, TH et al. [42] | 2019 | Facility based | " | Addis Abeba | Urban | Health care professionals | 19–65 | 422 | 77.9 | 9 |
| Zeru, Y et al. [43] | 2019 | Facility based | " | Addis Abeba | Urban | HEW | Not clearly stated | 508 | 79.2 | 8 |
| Degefu D et al. [68] | 2019 | Facility based | " | Gambella town, Gambela | Urban | Health care professionals | Not clearly stated | 161 | 61.5 | 9 |
| Tura T et al. [59] | 2019 | Institution based (school) | " | Wolayita, SNNPR | Urban | Students | Not clearly stated | 218 | 83.4 | 9 |

Key: CR:-Cross-sectional, HEW:—health extension workers, RAW-: reproductive age women, SNNPR:- Southern Nations, Nationalities, and People's Region.

## Prevalence of breast self-examination practice in Ethiopia

The lowest prevalence of breast self-examination practice was 3.6% observed in a study conducted in Oromia region [45] and the highest was 83.4% observed in a study conducted in SNNP region [59]. The $I^2$ (variation in ES attributable to heterogeneity) test result showed that there was high heterogeneity with $I^2$ = 99.3, at p-value <0.001. The pooled national level of breast self-examination practice was 36.72% (95% CI: 29.9%, 43.53%) based on the random effect analysis (Fig 2). Therefore, a meta-regression was conducted based on the year of publications, region and sample size to identify the source of heterogeneity. The result of meta regressions revealed that year of publication and regions of the study were found to be statistically significant (Table 2). The funnel plot asymmetry was used to check the presence of publication bias, and it showed that there was symmetrical distribution (Fig 3). Thus, to confirm this asymmetry, we conducted an Egger's and Begg's tests. The results of Egger's and Begg's tests showed that there was statistically significant publication bias in estimating the practice of breast self-examination among women in Ethiopia [(*p-value = 0.001*) and (*p-value = 0.043*)] respectively. Therefore, to account for publication bias, the duval and trimmed filled analysis was done.

A subgroup analysis by region and study population was also calculated to compare breast self-examination practice across regions of the country and among different study populations

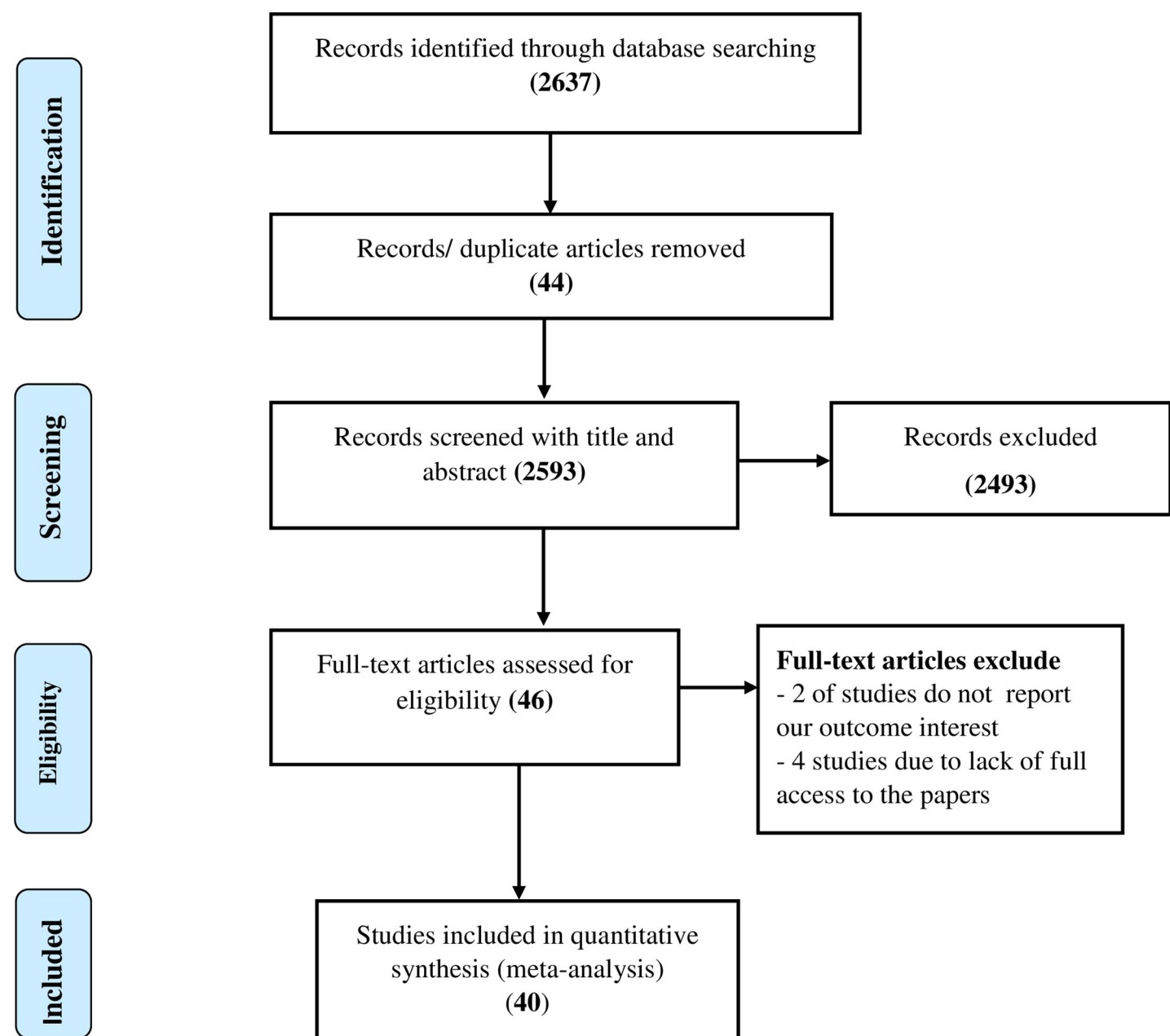

**Fig 1. Flow diagram of the included studies in the systematic review and meta- analysis of breast self-examination practice among women in Ethiopia.**

included. Accordingly, the highest practice of breast self-examination was observed in Gambela, 61.5% (95% CI: 53.98, 69.02) followed by Amhara region, 40.5% (95% CI: 33.31, 47.69) and Addis Ababa 42.52% (95% CI: 27.99, 57.05). Whereas, the lowest practice, 23% (95%CI: 15.65, 30.35), was observed in the Harari region.

The highest magnitude was observed among those studies which included non-reproductive age women 39.4 (22.01, 56.8). Moreover, study participating from urban areas had the highest 41.55 (33.97, 49.13) breast self-examination practice compared to semi urban regions 17.22 (11.0, 23.43) (Table 3).

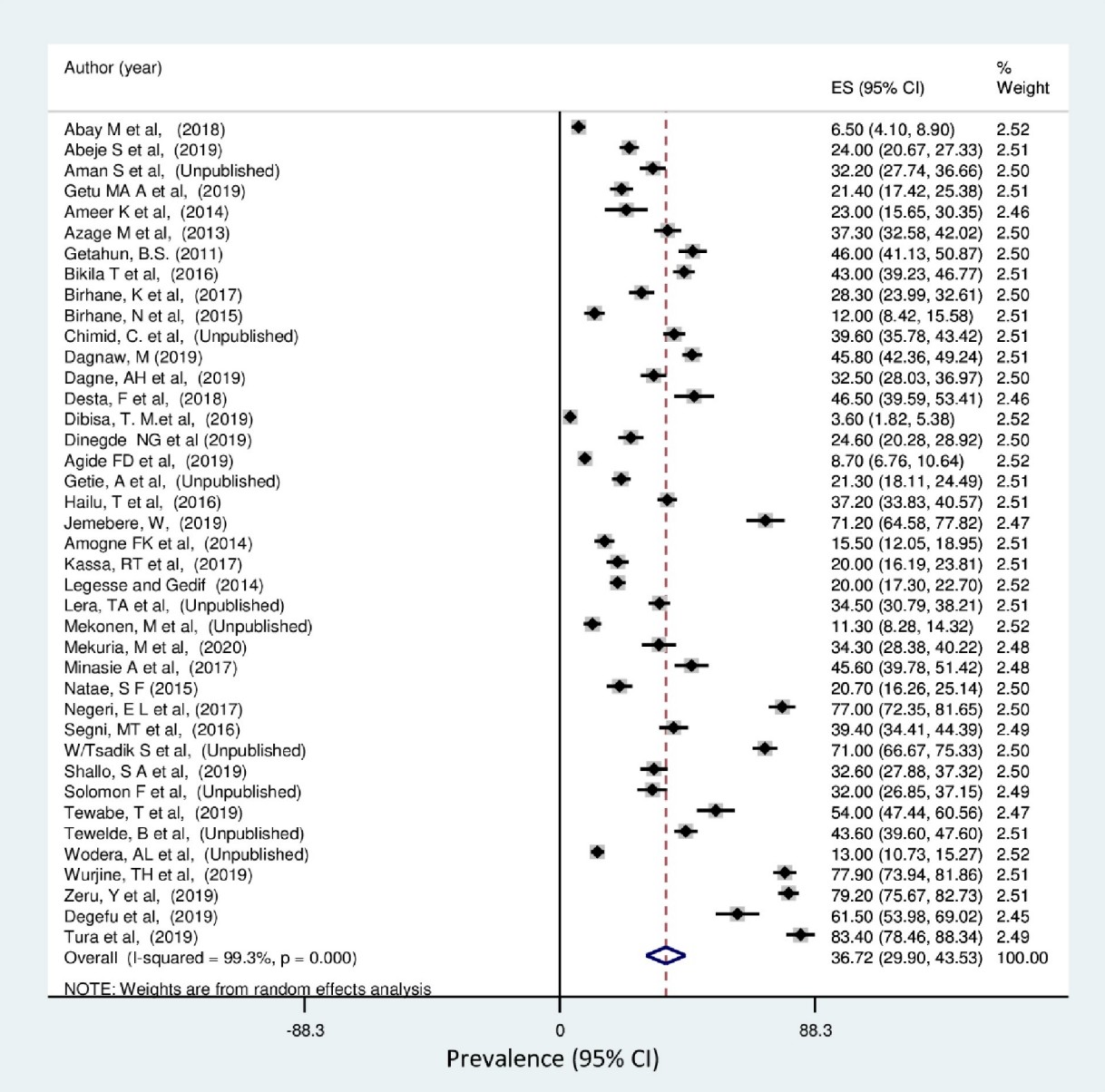

**Fig 2. Forest plot pooled practice of breast self-examination in Ethiopia: Meta-analysis.**

Additionally, the highest practice of breast self-examination was among health care workers 58.59% (95% CI: 45.69, 71.49) followed by among students, 35.87% (95% CI: 26.93, 44.81). While the lowest practice was observed among the general population 20.43% (95% CI: 14.13, 26.72) (Fig 4).

**Table 2. Meta-regression output to explore the source of heterogeneity of the pooled prevalence of breast self-examination practice among women in Ethiopia.**

| Variables | Coefficients | P-value | 95% conf. Interval | |
|---|---|---|---|---|
| Publication year | 42.4914 | <0.001 | 20.94471 | 64.03809 |
| Study population | 13.93318 | 0.066 | -.9874535 | 28.85381 |
| Sample size | 52.13567 | 0.637 | -169.7079 | 273.9792 |
| Regions | 43.07335 | <0.001 | 29.41824 | 56.72845 |

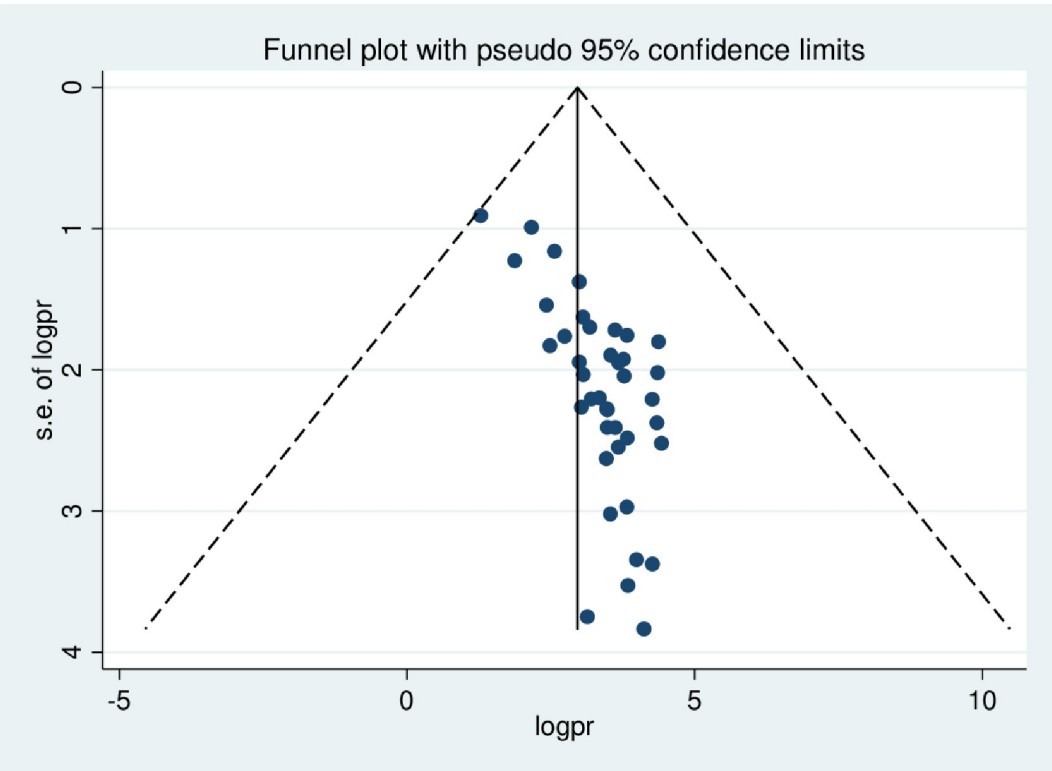

**Fig 3. The funnel plot asymmetry for publication bias of the studies included in systematic review and meta-analysis of breast self-examination practice among women in Ethiopia.**

## Determinants of breast self-examination practice in Ethiopia

Non-formal educational status of women OR = 0.4 (95% CI: 0.21, 0.77), family history of breast cancer OR = 2.04 (95% CI: 1.23, 3.39), good knowledge towards breast self-examination OR = 4.8 (95% CI: 3.03, 7.60]) and favorable attitude toward breast self-examination

**Table 3. Subgroup analysis of practice of breast self-examination by different study characteristics in Ethiopia.**

| By Region | Number of included studies | Prevalence (95%CI) | P-value | $I^2$ |
|---|---|---|---|---|
| Addis Ababa | 11 | 42.52 (27.99, 57.05) | < 0.001 | 99.18 |
| Oromia | 09 | 29.71 (16.52, 42.89) | < 0.001 | 99.15 |
| SNNPR | 08 | 38.7 (21.70, 55.85) | < 0.001 | 99.38 |
| Amhara | 06 | 40.5 (33.31, 47.69) | < 0.001 | 98.41 |
| Tigray | 03 | 21.2 (4.49, 37.91) | < 0.001 | 99.1 |
| Gambella | 01 | 61.5 (53.98, 69.02) | - | - |
| Dire Dawa | 01 | 32.00 (26.85, 37.15) | - | - |
| Harari | 01 | 23.00 (15.65, 30.35) | - | - |
| **By age of respondents** | **Number of included studies** | **Prevalence (95%CI)** | **P-value** | $I^2$ |
| Non reproductive age (>49 years) | 09 | 39.4 (22.01, 56.8) | <0.001 | 99.5 |
| Reproductive age | 12 | 36.87 (25.94, 47.81) | <0.001 | 99.1 |
| **By residency** | **Number of included studies** | **Prevalence (95%CI)** | **P-value** | $I^2$ |
| Urban | 32 | 41.55 (33.97, 49.13) | <0.001 | 99.1 |
| Urban and rural | 8 | 17.22 (11.0, 23.43) | <0.001 | 97.5 |

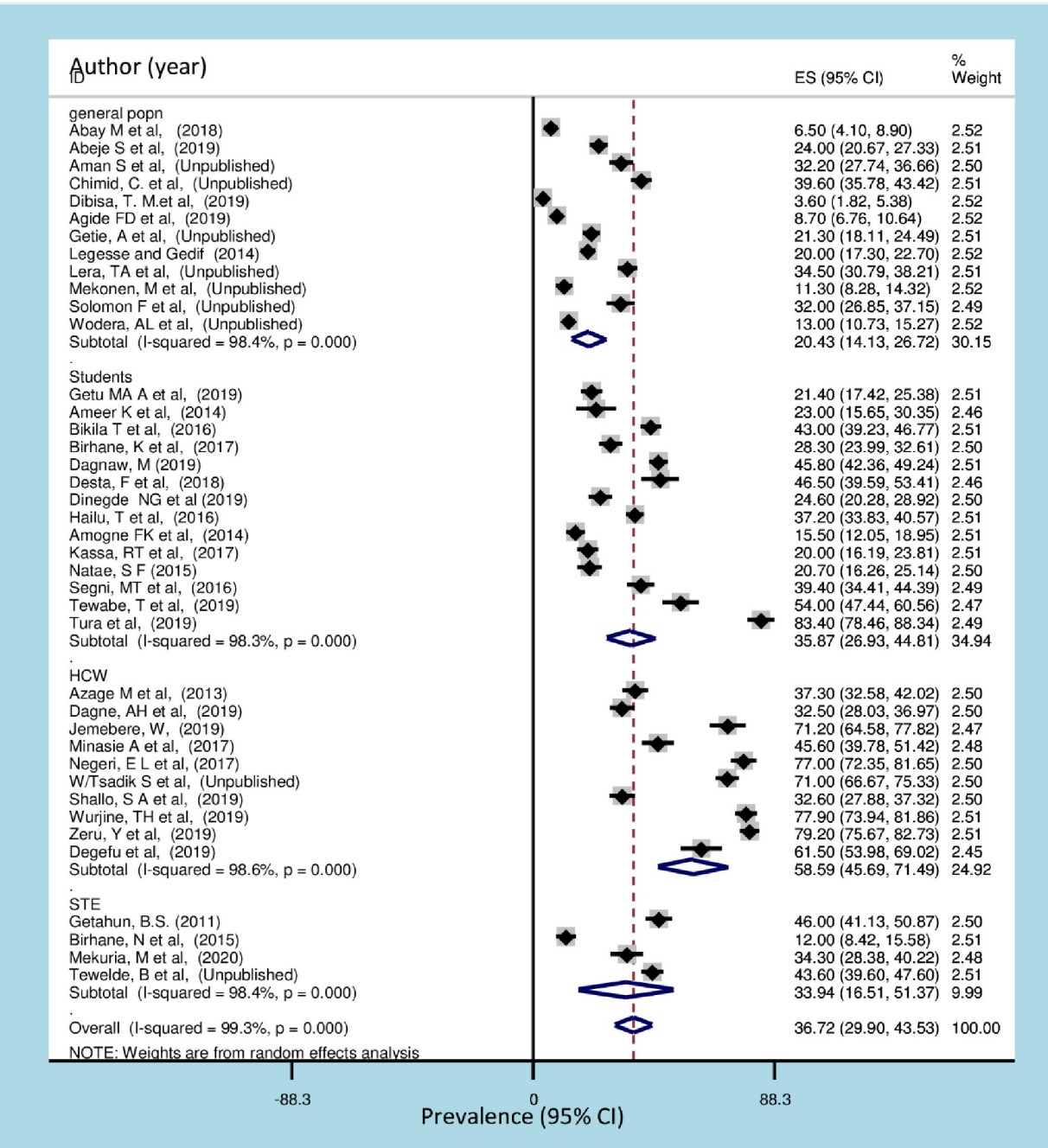

**Fig 4. Forest plot of subgroup analysis of breast self-examination among different populations of women in Ethiopia.** HCW: health care worker STE: school teacher.

OR = 2.75, [95% CI: 1.66, 4.55)] were significantly associated with practice of breast self-examination.

Seven studies were assessed for the association between educational status and breast self-examination practice. Accordingly, women with lower level of education were 60% less likely to practice breast self-examination than those who had higher level of education (OR = 0.40 [95% CI: 0.21, 0.77]) (Fig 5).

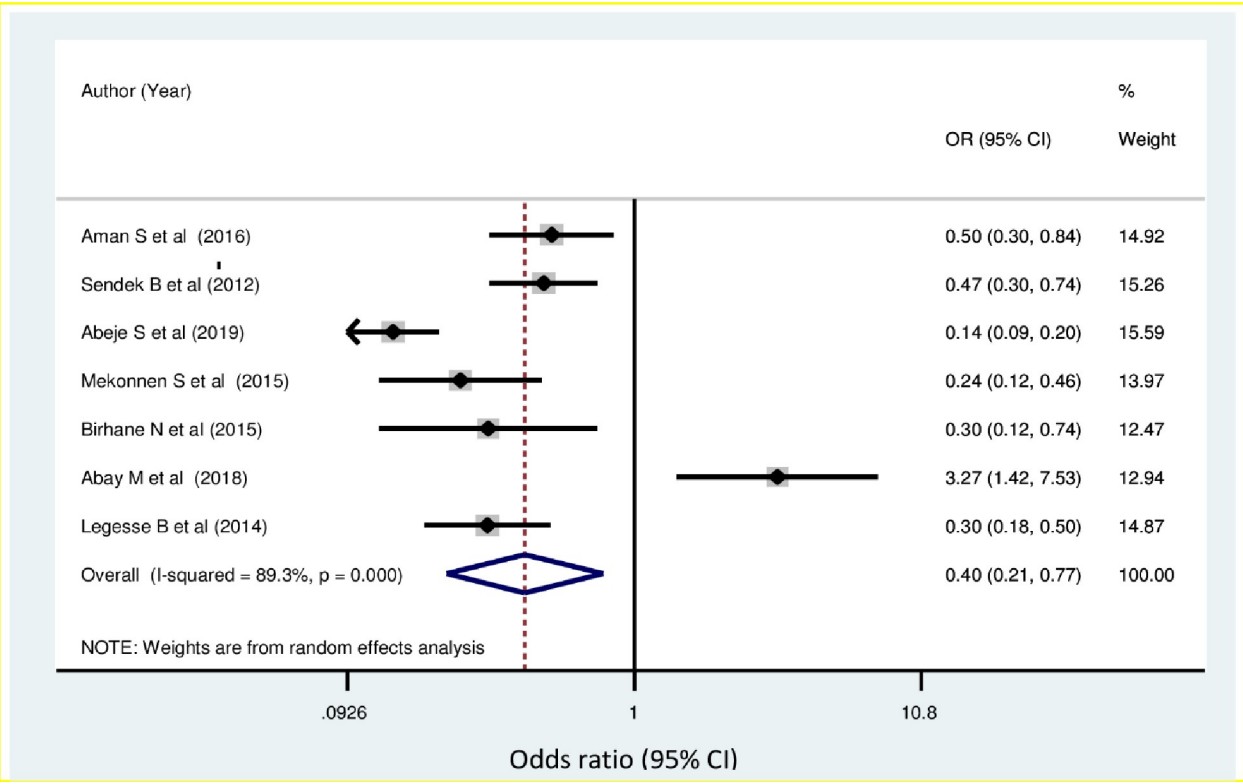

**Fig 5. Forest plot showing pooled odds ratio (log scale) of the associations between practice of breast self-examination among women's and educational status of the women.**

In the current study, the odds of breast self-examination use among women who have family history of breast cancer were about 2.04 times higher than women who have no family history of breast cancer (OR = 2.04 [95% CI: 1.23, 3.39]) (Fig 6).

Women who had good knowledge of breast self-examination were 4.8 times more likely to practice breast self-examination when compared with those women who had poor knowledge of breast self-examination (OR = 4.8, [95% CI: 3.03, 7.60]) (Fig 7).

Women who had a favorable attitude towards breast self-examination were 2.75 more likely to practice breast self-examination than those with unfavorable attitude OR = 2.75, [95% CI: 1.66, 4.55)] (Fig 8).

## Discussion

There exists some variation among different studies conducted on the effectiveness of breast self-examination. A meta-analysis conducted by Ku, YL to determine the value of breast self-examination suggested a positive relationship between BSE behaviour and stage of breast cancer at diagnosis [72]. The review reported a significant relationship between BSE behavior and survival rates; however, five studies reported no significant relationship between BSE and survival. In addition, a follow-up study from Thailand revealed that a significantly higher proportion of smaller tumor size, earlier stage, and better survival rates in women who regularly practiced BSE versus women who did not practice BSE [73].

Breast self-examination is a low-cost, low risk procedure that can be repeated at frequent intervals, and has been advocated as a self-performed screening procedure. Women who practice breast self-examination have a better survival [74]. Breast self-examination also results in

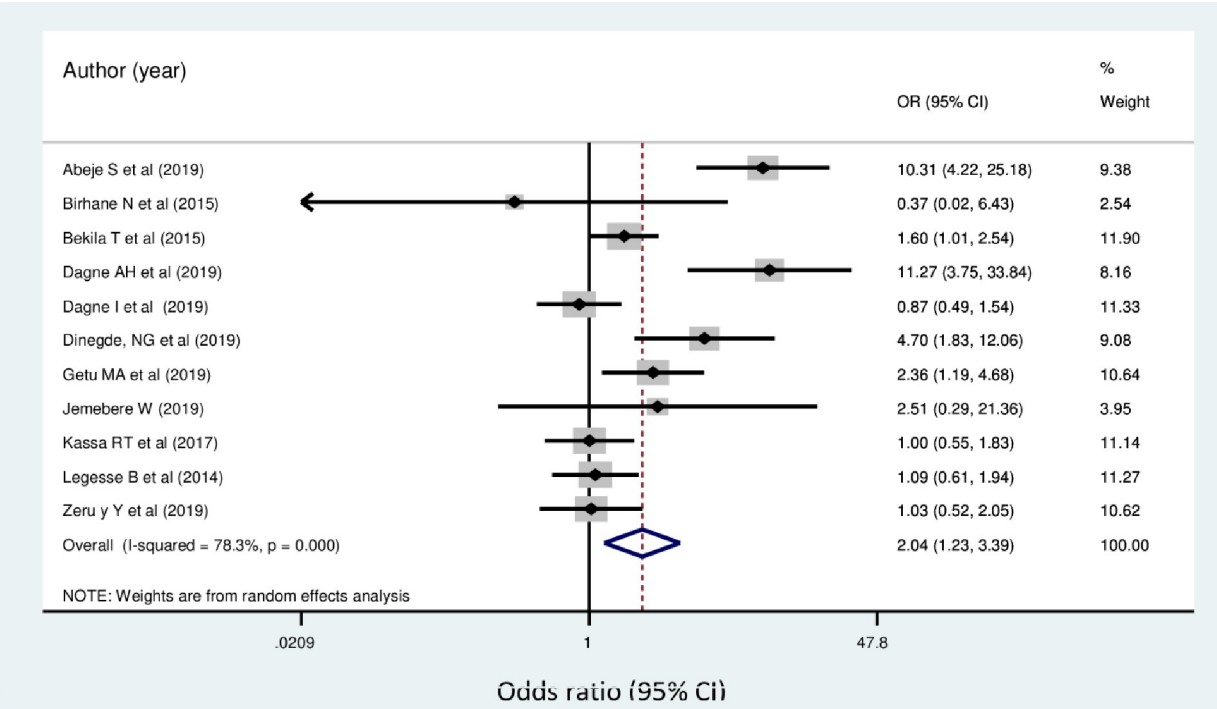

**Fig 6. Forest plot showing pooled odds ratio (log scale) of the associations between practice of breast self-examination among women's and family history of breast cancer.**

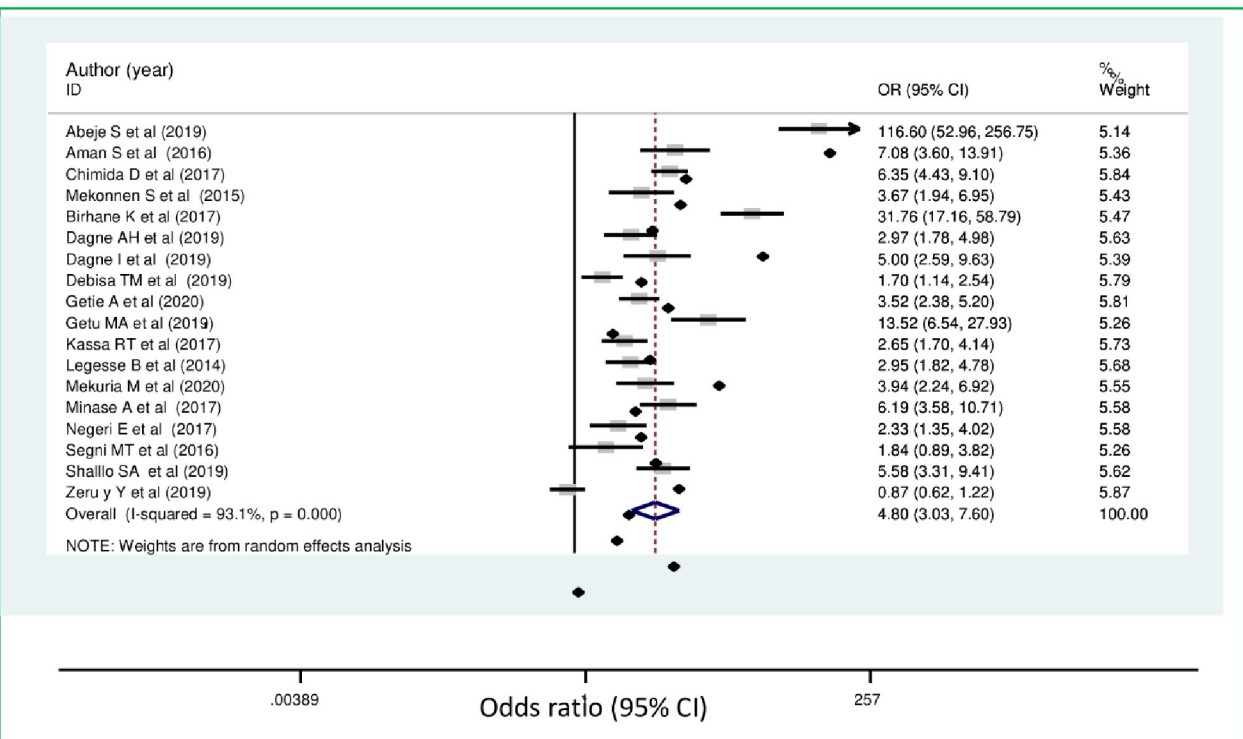

**Fig 7. Forest plot showing pooled odds ratio (log scale) of the associations between practice of breast self-examination among women's and knowledge of breast cancer.**

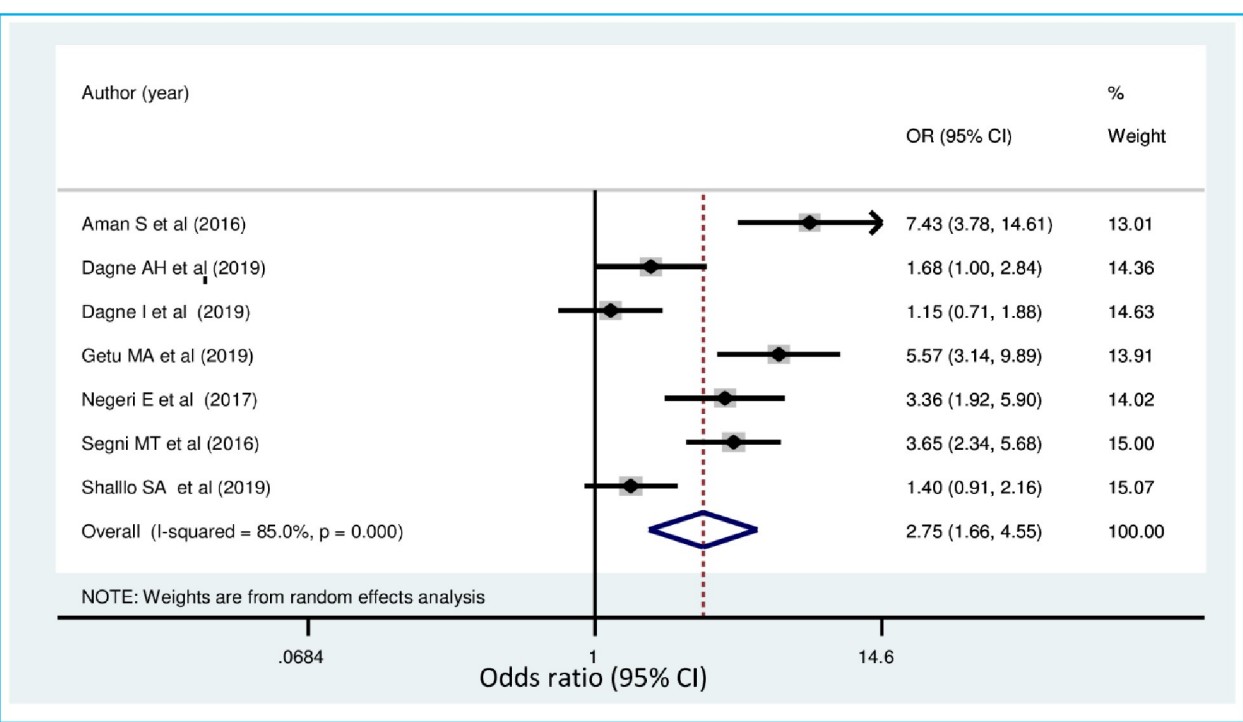

**Fig 8. Forest plot showing pooled odds ratio (log scale) of the associations between practice of breast self-examination among women's and attitude towards breast cancer.**

self-discovered cancers at an early stage at diagnosis in women who report practicing systematic BSE in contrast to those who do not [75]. In Ethiopia, studies have revealed that lower survival rate after late diagnosis, where nearly half of the patients die within 2 years [76, 77]. The current review revealed significant proportion of women do not practice breast examination.

This systematic review and meta-analysis is significant for understanding the existing practices of breast self-examination and its associated factors. Our findings map the existing evidence and summarize the findings as presented across the studies on promoting prevention and the early detection of breast cancer for women in Ethiopia. Breast self-examination is one of the active interventions that has shown to decrease incidence, late presentation and death related to breast cancer among women [78]. In resource limited countries like Ethiopia where expensive screening process like mammography is not readily available, breast self-examination is one of the cheapest and accessible method for screening breast cancer [79, 80].

A total of 40 articles with 17,820 women were included in the final analysis. We found the practice of breast self-examination to be 36.72% (95% CI: 29.9, 43.53). The current practice of breast self-examination estimate was in line with studies reported from Uganda, 30% [81], Nigeria, 39% [82], Ghana 32% [83], Pakistan 33% [84], Malaysia 36.7% [85], Iran 38.4%, 31.7% [86, 87] and Vietnam 39.9% [88].

On the other hand, the prevalence of breast self-examination practice used observed in the present study was lower than the prevalence reported that range between 54.2% to 60% [89–91] in Sub-Saharan African countries, between 46.3 to 87.4% [92–99] in different middle east countries, between 44.4% to 57.4% [10, 100–102], in Asian countries, 46% to 69.6% [103–105] in European countries and 93% in Australia [106]. This could be due to the difference in educational awareness creation on breast cancer and breast self-examination in the most parts of the developed world and the existence of supportive services for enhancement of breast self-

examination in those countries. The other possible explanation could be due to poor health seeking behavior, even though the majority of diseases including breast cancer affecting the populations of developing countries are preventable, the poor health seeking behavior due to different conditions affects the practice and the outcome of the disease [107]. Moreover, the socio-cultural difference among women in different parts of the world could also explain the existing discrepancy. Tetteh, DA et al. stated that in sub-Saharan African countries, cultural beliefs about the cause of breast diseases and associations of the female breasts with nurturance, motherhood and femininity influence the early diagnosis and management of breast cancer. Some cultural beliefs and understanding affect how diseases are diagnosed and treated, steps taken to manage the disease and ultimately how the disease is experienced [108]. A significant proportion of the study participants didn't practice breast self-examination. The increasing prevalence and morbidities from breast cancer among women worldwide particularly in developing countries including Ethiopia, where the majority of cases are diagnosed in late stages [9, 14] shows a need for the creation of nationwide awareness and population based breast self-examination programs. Different studies showed that the effect of breast self-examination education on breast self-examination practice [109, 110] and mortality related to breast cancer [11, 111]. An experimental study conducted in Iraq to determine the impact of education program on breast self-examination revealed that providing breast self-examination education have a positive impact on knowledge and practices of breast self-examination [112].

However, the observed magnitude of breast self-examination practice in this review was greater than the research findings in Eretria 13.4% [113], Nigeria ranged between 18.1% to 24.8% [114–116], Cameroon 4.1% [117], Tanzania 18.5% [118], Egypt 7.9% [119], Botswana 23.5% [120], Turkey ranging between 12.6% - 27% [121, 122], Yemen 17.4% [123], Iran 26% [124] Malaysia 25.5% [125], Vietnam 13.8% [126] and systematic review in Iran 21.9% [127]. The observed variation might be due to the different population groups (students, health care workers, general population) which are included in the current review.

The subgroup analysis by region was calculated to compare the practice of breast self-examination across regions of the country. Accordingly, the highest practice of breast self-examination was observed in Gambela followed by Amhara region and Addis Ababa. This could be due to different population group included in the study (health care workers were the participants) and the number of studies retrieved from each region, 11 studies from Addis Ababa. Moreover, difference in socio-demographic characteristics, the lifestyle activities and improved health seeking behavior in Addis Ababa and Tigray region might be associated with socioeconomic differences that might result in the practice of breast self-examination. This raises the alarm for other regions in committing to improve the most efficient and cost-effective approach, breast self-examination practice. The late stage at diagnosis coupled with a lack of comprehensive treatment centers leads to poor prognoses for women with breast cancer in sub-Sharan African countries and contributes to high mortality rates from this disease [128]. Although advanced screening techniques like mammography has led to mortality reduction in many high-income countries, it is currently not available in most countries of Sub Sharan African including Ethiopia. The oncology services are wholly inadequate and the country is not well-prepared to bear the growing burden of cancer and may not be the most appropriate early detection method for this region [25, 128]. Advocating the practice of breast self-examination in all regions of the country should be considered.

The practice of breast self-examination was compared by calculating subgroup analysis across different population groups included in the review. The highest prevalence was observed among health care workers and students. This could be due to the fact that both the health care providers and students throughout the journey of their profession had some way of being exposed to the experience of breast self-examination. The lowest prevalence was

observed in the general population, this suggests that health promotion and advocacy should be emphasized at the community and population level. In resource limited countries, it is recommended to develop culturally sensitive, linguistically appropriate local education programs for target populations to teach value of early detection, breast cancer risk factors and breast health awareness addressing both the education and breast health awareness [129]. The so far promising achievement of health extension program in Ethiopia [130, 131] should one of the mechanism through which awareness and knowledge disseminated about breast cancer. In low and middle income settings where there is shortage of health workers' to provide early detection services, community health workers' (CHWs), have been proposed in order to achieve universal health coverage [132].

The sub group analysis computed among different age group showed that the highest prevalence was observed among those studies which included non-reproductive age women. This finding is supported by studies from Indonesia [10] and Sweden [105]. This could be due to the high risk of exposure to breast cancer for women above the age of 50 [133]. However, in both the reproductive and non-reproductive age categories significant proportion of women, 63% and 60.7 respectively do not have the practice of breast self-examination. Educational programs on breast self-examination [110] should be established at national level targeting different age groups.

Moreover, study from urban areas had the highest breast self-examination practice compared to semi urban regions. This could be due accessibility and exposure to health related behaviours varies among the urban and rural population. Center for disease control and prevention (CDC) on the morbidity and mortality weekly report classified rural residents as health disparity populations [134]. National breast self-examination educations should address health disparities associated with rural residence.

In this review, women with non-formal education were less likely to practice breast self-examination than those who had a higher level of education. This finding is supported by other studies conducted in Nigeria [115, 135, 136], Ghana [83, 90], Indonesia [10], Iran [94, 137], Beirut [98] and Vietnam [88]. This could be explained by the fact the more educated women may probably have be exposed to different health awareness information. Educated women might tend to have more ways of exploring the opportunities and experiences around, and this is supported by Sani et al., who mentioned that education plays an important role in behavior modification and may lead to cues to action [49]. A study conducted in middle- and low-income countries also indicated a significant relationship between educational level and breast self-examination practice of BSE [135, 138].

This meta-analysis also revealed that women who had good knowledge of breast self-examination had greater odds of breast self-examination practice than those women who had poor knowledge. The finding is supported by a study done in Nigeria [135, 139], Cameroon [140], Ghana [141], Malaysia [142], Turkey [122], Saudi [143], Iran [94, 124], Iraq [144], Vietnam [126] and Italy [104]. The possible explanation could be the knowledge of breast self-examination provided women an opportunity and experience to become aware and familiar with the normal shape and feeling of their breast. Seni et al. stated that Knowledge of breast self-examination is important in the detection and diagnosis of breast cancer; more than 80% of the cancer patients detect their own tumor [139].

Moreover, women who had a family history of breast cancer were more likely to practice breast self-examination than those who had no family history of breast cancer. This finding is supported by the studies done in Indonesia [10], Saudi [143], Malaysia [102] and Turkey [96]. This could be due to the fear they develop from the experience of the family members, which could lead to perceived susceptibility and cancer worries that in turn positively affects the level of participation of breast self-examination [145].

Women who have favorable attitude towards breast self-examination were more likely to practice breast self-examination than those with unfavorable attitude. This finding is supported by the studies done in Cameroon [140], Ghana [90] and Italy [104]. This might be due to the fact that women with positive attitude about breast examination tend to have better understanding of BSE and thereby practice breast self-examination [146].

## Strengths and limitations of the study

The strength of this meta-analysis is that it is the first of its kind in Ethiopia and it lays ground in the journey for existing and unpublished research. In addition, majority of the region in Ethiopia are represented in this review, which strengthens its representativeness. Nevertheless, the results of this systematic review and meta-analysis should cautiously be interpreted. The high heterogeneity of results among studies may be explained by difference in the characteristics of the studies, setting, and this may have led to insufficient statistical power to detect significant association. Accordingly, a meta-regression analysis revealed that there was variation due to region and publication year. Furthermore, all the included studies were cross-sectional which couldn't show temporal relationship between the predictors and practice of breast self-examination.

## Conclusions

The level of breast self-examination practice among women in Ethiopia was found significantly low noting that all females of reproductive age are expected to practice breast self-examination. Knowledge about breast self-examination, educational status, attitude towards breast self-examination and family history of breast cancer were the predictor variables. From the sub group analysis, the lowest practice of breast self-examination was among the general population. Hence, the Ethiopian ministry of health, Regional health bureaus, health facilities, higher education centers and other stakeholders should consider the identified factors and, therefore, broader approaches to reach the broader population to enhance the habit of breast self-examination. Because of the expensive and inaccessibility of the advanced screening mechanisms for early detection of breast cancer, it would be inclusive of the majority to invest on breast self-examination to combat the burden of breast cancer in the country.

## Supporting information

**S1 Checklist. Reporting items for systematic review and meta-analysis (PRISMA-P) 2009 statement guideline.**
(DOC)

**S1 File. Search strategy used for the systematic and meta-analysis on breast self-examination practice and its determinants among women in Ethiopia: A systematic review and meta-analysis.**
(DOC)

**S2 File. JBI critical appraisal checklist for studies reporting prevalence data.**
(XLSX)

## Acknowledgments

We would like to acknowledge to authors of studies included in this review.

## Author Contributions

**Conceptualization:** Yordanos Gizachew Yeshitila, Melaku Desta.

**Data curation:** Yordanos Gizachew Yeshitila.

**Formal analysis:** Yordanos Gizachew Yeshitila.

**Funding acquisition:** Yordanos Gizachew Yeshitila.

**Investigation:** Yordanos Gizachew Yeshitila.

**Methodology:** Yordanos Gizachew Yeshitila, Melaku Desta.

**Project administration:** Yordanos Gizachew Yeshitila.

**Resources:** Yordanos Gizachew Yeshitila, Melaku Desta.

**Software:** Yordanos Gizachew Yeshitila, Melaku Desta.

**Supervision:** Yordanos Gizachew Yeshitila, Getachew Mullu Kassa, Melaku Desta.

**Validation:** Yordanos Gizachew Yeshitila, Getachew Mullu Kassa, Selamawit Gebeyehu, Peter Memiah, Melaku Desta.

**Visualization:** Yordanos Gizachew Yeshitila, Getachew Mullu Kassa, Selamawit Gebeyehu, Peter Memiah, Melaku Desta.

**Writing – original draft:** Yordanos Gizachew Yeshitila, Getachew Mullu Kassa, Selamawit Gebeyehu, Melaku Desta.

**Writing – review & editing:** Yordanos Gizachew Yeshitila, Getachew Mullu Kassa, Selamawit Gebeyehu, Peter Memiah, Melaku Desta.

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
