## [Decision Letter · Decision Letter 0]

21 Oct 2020

PONE-D-20-25679

Breast self-examination practice and its determinants among women in Ethiopia: a systematic review and meta-analysis

PLOS ONE

Dear Dr. Yeshitila,

Thank you for submitting your manuscript to PLOS ONE. After careful consideration, we feel that it has merit but does not fully meet PLOS ONE’s publication criteria as it currently stands. Therefore, we invite you to submit a revised version of the manuscript that addresses the points raised during the review process.

Please address all comments and concerns  made by the reviewers during review process.I am particularly interested in your response to "methods used in statistical analysis" query.

We look forward to receiving your revised manuscript.

Kind regards,

Amir Radfar, MD,MPH,MSc,DHSc

Academic Editor

PLOS ONE

Journal Requirements:

2. Please attach a Supplemental file of the results of the individual components of the quality assessment, not just the overall score, for each study included. Please also explain the reasons, and number of studies excluded for each reason, in the flow diagram. Thank you.

3. Thank you for including the statement that "The primary online search for review was conducted between February 16 to May 21 2020.". Please revise this statement to clarify whether all databases were searched from inception, or if there were any limits placed on the publication dates in your search.

5. We note that this manuscript is a systematic review or meta-analysis; our author guidelines therefore require that you use PRISMA guidance to help improve reporting quality of this type of study. Please upload copies of the completed PRISMA checklist as Supporting Information with a file name “PRISMA checklist”.

Reviewers' comments:

Reviewer's Responses to Questions

**Comments to the Author**

1. Is the manuscript technically sound, and do the data support the conclusions?

Reviewer #1: Partly

Reviewer #2: Yes

2. Has the statistical analysis been performed appropriately and rigorously? 

Reviewer #1: No

Reviewer #2: I Don't Know

3. Have the authors made all data underlying the findings in their manuscript fully available?

Reviewer #1: Yes

Reviewer #2: Yes

4. Is the manuscript presented in an intelligible fashion and written in standard English?

Reviewer #1: Yes

Reviewer #2: No

5. Review Comments to the Author

Reviewer #1: This paper describes a meta-analysis whose purpose is to examine in Ethiopia compliance with the WHO recommendation that women of reproductive age practice breast self-examination (BSE), and to identify factors association with variation in BSE practice. This information is obviously of potential value in planning educational activities, and does represent original research.

The search strategy, sites searched and search terms used, are described and the eligible papers (42 of 2637) identified by consensus.

Table 2 provides some information about the papers examined, the populations studied in them and the prevalence of BSE in each. It is concluded that only about a third of women in Ethiopia comply with the WHO recommendation re BSE.

Areas of concern:

1. Data analyzed: Table 2 shows some characteristics of the studies included. Some include subjects to age 80, and others “older” subjects of unspecified age.

These studies do not contribute to compliance with the WHO recommendation directed at women of reproductive age. If the results of these studies cannot be examined in relation to age I would suggest omitting them, or at least examining the effect of omitting them on the overall conclusion.

The sampling methods used by the included studies are not clear. Did any use random sampling in a defined population?

It would also be useful to indicate in Table 2 which of the regions of Ethiopia included are rural or urban.

There is no definition of compliance with BSE- can a distinction be made between never, occasional and regular use? Examples of the wording of questions used by studies re BSE would be useful.

2. Analysis of determinants of BSE: The analysis of factors associated with practice of BSE appears to be entirely univariable. Variables such as educational level, occupation, knowledge of breast cancer etc. may well be related. Multivariable analysis would identify which of the candidate variables are independently associated with BSE practice.

3. The discussion does not include any account of variation in opinions and data concerned with the usefulness of BSE and the effects of BSE on mortality from breast cancer, although most of this literature comes from Europe of North America.

Reviewer #2: In answering Question 4, the manuscript was written in standard English, but there were a number of points needing clarification.

- Articles were often not used, as in the Abstract, where it starts "Survival rate from ..." when it should say "The survival rate...."

- The number of subjects should be given in the Abstract as well as the number of articles included. Also, it gives the lowest percentage of breast self-examination as being "...among the general population".; What does this mean, an average of the country? In the conclusion, it is stated "Understanding the syndemic factors..." I am not sure "syndemic" is the proper word to use here.

- In the Introduction, it says that breast self-examination has been promoted as a screening method. In the fourth paragraph, line 83, it states that "BSA is recommended for raising awareness ....rather than a screening method." BSA stands for breast self-awareness; do you actually mean BSE (breast self-exam)?

- In the next paragraph, it states "The practice of breast self-examination varies across Ethiopia with different variations across women population." Do you mean across the population of women?

- In Table 1, Study Populations of Students, Employees, and Teachers are given with no age ranges or other details. Have these been omitted, or did the studies used not have this information? Below Table 1 in Prevalence of Breast self-examination practices in Ethiopia, a P value of 0.000 is given. Is this possible, or was a number left out?

- In the Discussion section, possibly paragraph 3, lines 359 and 360, "...calls for population based breast self-examination nationwide awareness creation programs...." Do you mean "...show a need for the creation of nationwide awareness and population based breast self-examination programs"? Two paragraphs below, line 371, it says "...(health care workers where the study participants)" Do you mean health care workers were the participants?

6. PLOS authors have the option to publish the peer review history of their article (what does this mean?). If published, this will include your full peer review and any attached files.

Reviewer #1: **Yes: **Dr Norman Boyd

Reviewer #2: No

---

## [Author Response · Author response to Decision Letter 0]

12 Nov 2020

Responses to editor’s comment 

Editor’s comments, suggestions and question are considered and carefully revised the manuscript as per the suggestions and comments. I am very thankful to the potential editor for suggestions and comments, which substantially improved the manuscript

https://journals.plos.org/plosone/s/file?id=ba62/PLOSOne_formatting_sample_title_authors_affiliations.pdf Thank you, dear editor, we have re-written the manuscript based on PLOS ONE's style requirements

Please attach a Supplemental file of the results of the individual components of the quality assessment, not just the overall score, for each study included. Please also explain the reasons, and number of studies excluded for each reason, in the flow diagram. Thank you. Thank you, dear editor, we have attached supplementary file of the results of the individual components of the quality assessment in the current version of the revised manuscript (attached as supplementary file 3, Microsoft excel format) 

Thank you for including the statement that "The primary online search for review was conducted between February 16 to May 21 2020.” Please revise this statement to clarify whether all databases were searched from inception, or if there were any limits placed on the publication dates in your search. Thank you dear editor for the meticulous observation, we have included studies irrespective of their publication date through May 21 2020 (mentioned in the manuscript in method section, under sub heading Data extraction and quality assessment, Line 163, Page 8)

Please include captions for your Supporting Information files at the end of your manuscript, and update any in-text citations to match accordingly. Please see our Supporting Information guidelines for more

 information: http://journals.plos.org/plosone/s/supporting-information. We have attached the following supporting information at end of the manuscript, 

• PRISMA checklist: Reporting Items for Systematic Review and Meta-Analysis (PRISMA-P) 2009 statement guideline

• Supplementary file 2: search strategy used for the systematic and Meta-analysis on Breast self-examination practice and its determinants among women in Ethiopia: a systematic review and meta-analysis

• Supplementary file 3: JBI Critical Appraisal Checklist for Studies Reporting Prevalence Data

We note that this manuscript is a systematic review or meta-analysis; our author guidelines therefore require that you use PRISMA guidance to help improve reporting quality of this type of study. Please upload copies of the completed PRISMA checklist as Supporting Information with a file name “PRISMA checklist” We have uploaded copies of the completed PRISMA checklist as Supporting Information with a file name “PRISMA checklist

Responses to the Reviewers' comment and question 

Reviewer’s comments are considered and carefully revised the manuscript as per the suggestions and comments. I am very thankful to the potential reviewers for suggestions and comments, which substantially improved the manuscript

Reviewer 1 comment and question Responses Reviewer 1

Data analyzed: Table 2 shows some characteristics of the studies included. Some include subjects to age 80, and others “older” subjects of unspecified age.

These studies do not contribute to compliance with the WHO recommendation directed at women of reproductive age. If the results of these studies cannot be examined in relation to age I would suggest omitting them, or at least examining the effect of omitting them on the overall conclusion. 1. We have included the age of the respondents under the Result section , sub heading Study identification, table 1 (Page 11-13)

2. Dear reviewer based on the references we found and our best understanding, women above reproductive age group should be considered for breast self-examination especially in resource limited countries and most of the studies included under the current review included non-reproductive age group (WHO, and other organizations recommended the starting age ( starting from age of 20 where the breast develops or matures) of breast self-examination, however the upper age wasn’t restricted or specified. A recommendation from WHO (1) shows that beginning at the age of 20, women should examine their breast every two or three years and, increasing one year from the age of 40. Studies shows that, the risk of breast cancer increases dramatically after menopause (2). In low resource counties including Ethiopia where mammography is not affordable, it is feasible to extend the breast self-examination practice beyond reproductive age. 

**Besides we tried to see the sub group analysis by categorizing the studies into reproductive age and non-reproductive age (

The sampling methods used by the included studies are not clear. Did any use random sampling in a defined population? We have assessed the sampling method as one of the criteria under the quality appraisal, in addition we have prepared a separate file named sampling method (from the total of 41 studies included, 39 (95% of them used random method, of which 10 (25.6% used systematic random sampling)

It would also be useful to indicate in Table 2 which of the regions of Ethiopia included are rural or urban. Thank you so much dear reviewer, we have included one additional column for residence under result section, sub heading study identification (page 11-13), 31 (80.4% of the included studies were from urban areas, the rest 19.6 are from urban and rural regions. Additionally we conducted sub group analysis to see the variation accounted due to residence (

There is no definition of compliance with BSE- can a distinction be made between never, occasional and regular use? Examples of the wording of questions used by studies re BSE would be useful. Dear reviewer thank you so much, we have included the definition of the outcome variable, Examples of the wording of questions used by studies and we also included the independent variable list too under the method section (Line 150 -160), Page 7 to 8

Analysis of determinants of BSE: The analysis of factors associated with practice of BSE appears to be entirely univariable. Variables such as educational level, occupation, knowledge of breast cancer etc. may well be related. Multivariable analysis would identify which of the candidate variables are independently associated with BSE practice. Dear reviewer we applied the univariable analysis based on our knowledge and understanding of the following concept: Price MJ et al, in Cochrane review stated that When review has several outcomes of interest , and the effect estimates for these outcomes may be correlated within primary studies, because the same patients provide data towards them. Standard univariate meta‐analysis (UVMA) methods do not account for this correlation. In recognition of this, multivariate meta‐analysis (MVMA) models have been developed. In addition it was emphasized that multivariate meta‐analysis is suggested especially where there are missing data for some outcomes or where post estimation modelling requires intervention effect estimates for multiple correlated outcomes (3). Which we believe do not apply to our current study. 

The discussion does not include any account of variation in opinions and data concerned with the usefulness of BSE and the effects of BSE on mortality from breast cancer, although most of this literature comes from Europe of North America. 

Dear reviewer, thank you for pointing out this very important point which we failed to address, we have included the points under discussion (Line 351-365)

Reviewer 2 comment and question Responses to Reviewer 2

Articles were often not used, as in the Abstract, where it starts "Survival rate from ..." when it should say "The survival rate...." Corrected as per the suggestion (line 26 of the abstract and 91 in introduction section )

The number of subjects should be given in the Abstract as well as the number of articles included. Also, it gives the lowest percentage of breast self-examination as being "...among the general population".; What does this mean, an average of the country? In the conclusion, it is stated "Understanding the syndemic factors..." I am not sure "syndemic" is the proper word to use here. • Thank you dear reviewer, we have included the number of subjects and number of articles in the Abstract section under result subheading, (Line 41, Page 2)

• Dear reviewer, in the current review we have four different population groups (health professional, students, teachers and the general population in the community), during the subgroup analysis the highest prevalence of breast self-examination was among health professionals and the lowest prevalence was among the general population in the current review)

• The We apologize for the improper word use, the word syndemic, it rephrased in the paragraph (Line 54 -55, Page 3 

In the Introduction, it says that breast self-examination has been promoted as a screening method. In the fourth paragraph, line 83, it states that "BSA is recommended for raising awareness ....rather than a screening method." BSA stands for breast self-awareness; do you actually mean BSE (breast self-exam)? Dear reviewer thank you so much for pointing that out, we have comprehended the paragraph in incorrect way, we apologize for that, and yes it was supposed to be BSE. We have corrected based on your corrections and modified the paragraph too (Line 88-90)

In the next paragraph, it states "The practice of breast self-examination varies across Ethiopia with different variations across women population." Do you mean across the population of women? Thank you so much for your directions, it is corrected as across the population of women (line 95)

In Table 1, Study Populations of Students, Employees, and Teachers are given with no age ranges or other details. Have these been omitted, or did the studies used not have this information? Below Table 1 in Prevalence of Breast self-examination practices in Ethiopia, a P value of 0.000 is given. Is this possible, or was a number left out? 1. We have added one additional column and included the age of respondents in table one (Page 11-13) 

2. Dear reviewer we have corrected it as p<0.001under result section, sub heading prevalence of breast self-examination practice in Ethiopia, line 252, page 14 (The level of statistical significance is expressed as a p-value between 0 and 1. Some statistical software like STAT sometimes gives p value .000 which is impossible and must be taken as p< .001)

In the Discussion section, possibly paragraph 3, lines 359 and 360, "...calls for population based breast self-examination nationwide awareness creation programs...." Do you mean "...show a need for the creation of nationwide awareness and population based breast self-examination programs"? Two paragraphs below, line 371, it says "... (health care workers where the study participants)" Do you mean health care workers were the participants? 1. Thank you for pointing out the idea, the paragraph is modified according to your suggestion as: shows a need for the creation of nationwide awareness and population based breast self-examination programs, we have also added literatures to support and strengthen the idea. (Line 399-404, Page 

2. Thank you very much for your meticulous observation to the errors we were unable to detect, corrected as per your observation as: yes health care workers were the participants (line 415)

Additional corrections made by the authors 

Authors' Contributions corrected as PLOS ONE requirement Line 503-517

Acknowledgment section was mistakenly repeated and, therefore, deleted from the manuscript. Deleted from line 496

On discussion section, additional emphasis has been made on the contribution of educational programs on breast self-examination practice. Line 396-401, Page 24

Subgroup analysis done accounting for variations in residence and age category and supported with discussion Result section, subheading, Prevalence of Breast self-examination practice in Ethiopia, Table 2, Page 17

Discussion, Line 443-555, Page 27-28

---

## [Decision Letter · Decision Letter 1]

4 Dec 2020

PONE-D-20-25679R1

Breast self-examination practice and its determinants among women in Ethiopia: a systematic review and meta-analysis

PLOS ONE

Dear Dr. Yeshitila,

Thank you for submitting your manuscript to PLOS ONE. After careful consideration, we feel that it has merit but does not fully meet PLOS ONE’s publication criteria as it currently stands. Therefore, we invite you to submit a revised version of the manuscript that addresses the points raised during the review process.

ACADEMIC EDITOR: Please address all comments made by the reviewers and clarify discussion section accordingly.

We look forward to receiving your revised manuscript.

Kind regards,

Amir Radfar, MD,MPH,MSc,DHSc

Academic Editor

PLOS ONE

Reviewers' comments:

Reviewer's Responses to Questions

**Comments to the Author**

1. If the authors have adequately addressed your comments raised in a previous round of review and you feel that this manuscript is now acceptable for publication, you may indicate that here to bypass the “Comments to the Author” section, enter your conflict of interest statement in the “Confidential to Editor” section, and submit your "Accept" recommendation.

Reviewer #1: All comments have been addressed

Reviewer #2: All comments have been addressed

2. Is the manuscript technically sound, and do the data support the conclusions?

Reviewer #1: Yes

Reviewer #2: (No Response)

3. Has the statistical analysis been performed appropriately and rigorously? 

Reviewer #1: Yes

Reviewer #2: (No Response)

4. Have the authors made all data underlying the findings in their manuscript fully available?

Reviewer #1: Yes

Reviewer #2: (No Response)

5. Is the manuscript presented in an intelligible fashion and written in standard English?

Reviewer #1: Yes

Reviewer #2: (No Response)

6. Review Comments to the Author

Reviewer #1: The horizontal axis is unlabelled in all of the figures. Without this the figures cannot be interpreted.

Reviewer #2: This manuscript could still use brushing up in its use of proper English; however, most of the small mistakes do not lead to misunderstanding of the information. However, the Discussion section in the revision needs some more clarification.

- I would end the second sentence of the Discussion at "...stage of breast cancer at diagnosis." Then "The review reported a significant relationship between BSE behavior and survival rates; however, five studies reported no significant relationship between BSE and survival. In addition, a follow-up study from Thailand revealed.... and better survival rates in women who regularly practiced BSE versus women who did not practice BSE." I believe this is what you meant.

- In the next paragraph, third sentence, "Breast self-examination also results in self-discovered cancers at an early stage at diagnosis in women who report practicing systematic BSE in contrast to those who do not."

I believe this will clarify what was a meant; if not, it still needs to be re-written to become more clear.

7. PLOS authors have the option to publish the peer review history of their article (what does this mean?). If published, this will include your full peer review and any attached files.

Reviewer #1: **Yes: **Dr Norman Boyd

Reviewer #2: No

---

## [Author Response · Author response to Decision Letter 1]

23 Dec 2020

Reviewer 1 :The horizontal axis is unlabelled in all of the figures. Without this the figures cannot be interpreted.

Response: Dear reviewer, we are grateful for the particular observation and your valuable suggestion. We have provided label for the horizontal axis for all the figures based on your recommendations. 

Reviewer 2: This manuscript could still use brushing up in its use of proper English; however, most of the small mistakes do not lead to misunderstanding of the information. However, the Discussion section in the revision needs some more clarification.

- I would end the second sentence of the Discussion at "...stage of breast cancer at diagnosis." Then "The review reported a significant relationship between BSE behavior and survival rates; however, five studies reported no significant relationship between BSE and survival. In addition, a follow-up study from Thailand revealed.... and better survival rates in women who regularly practiced BSE versus women who did not practice BSE." I believe this is what you meant.

 In the next paragraph, third sentence, "Breast self-examination also results in self-discovered cancers at an early stage at diagnosis in women who report practicing systematic BSE in contrast to those who do not."

I believe this will clarify what was a meant; if not, it still needs to be re-written to become clearer

Response: Dear reviewer, we thank you so much for the meticulous observation and your valuable suggestions. Your suggestions are what we meant to write, we have amended and modified the paragraphs based on your suggestion. 

(Line 353-357, Page 23 and line 360-362, Page 24)

---

## [Editor Report · Decision Letter 2]

26 Dec 2020

Breast self-examination practice and its determinants among women in Ethiopia: a systematic review and meta-analysis

PONE-D-20-25679R2

Dear Dr. Yeshitila,

We’re pleased to inform you that your manuscript has been judged scientifically suitable for publication and will be formally accepted for publication once it meets all outstanding technical requirements.

Kind regards,

Amir Radfar, MD,MPH,MSc,DHSc

Academic Editor

PLOS ONE
---

## [Editor Report · Acceptance letter]

30 Dec 2020

PONE-D-20-25679R2 

Breast self-examination practice and its determinants among women in Ethiopia: a systematic review and meta-analysis 

Dear Dr. Yeshitila:

I'm pleased to inform you that your manuscript has been deemed suitable for publication in PLOS ONE. Congratulations! Your manuscript is now with our production department. 

Kind regards, 

on behalf of

Dr. Amir Radfar 

Academic Editor

PLOS ONE